# Factors and Mechanisms That Influence Chromatin-Mediated Enhancer–Promoter Interactions and Transcriptional Regulation

**DOI:** 10.3390/cancers14215404

**Published:** 2022-11-02

**Authors:** Shinsuke Ito, Nando Dulal Das, Takashi Umehara, Haruhiko Koseki

**Affiliations:** 1Laboratory for Developmental Genetics, RIKEN Center for Integrative Medical Sciences, Yokohama 230-0045, Japan; 2Laboratory for Epigenetics Drug Discovery, RIKEN Center for Biosystems Dynamics Research, Yokohama 230-0045, Japan; 3Immune Regulation, Advanced Research Departments, Graduate School of Medicine, Chiba University, Chiba 260-8677, Japan

**Keywords:** gene expression, enhancer, promoter, histone modifications, liquid–liquid phase separation, cancer

## Abstract

**Simple Summary:**

The physical interactions between enhancers and promoters create chromatin conformations involved in gene regulation. In cancer cells, the chromatin conformations can be altered with uncontrolled deposition of histone marks resulting in varied gene expression. Although it is not entirely comprehensive how chromatin-mediated enhancer–promoter (E–P) interactions with various histone marks can affect gene expression, this proximity has been observed in multiple systems at multiple loci and is thought to be essential to control gene expression. In this review, we focus on emerging views of chromatin conformations associated with the E–P interactions and factors that establish or maintain such interactions, which may regulate gene expression.

**Abstract:**

Eukaryotic gene expression is regulated through chromatin conformation, in which enhancers and promoters physically interact (E–P interactions). How such chromatin-mediated E–P interactions affect gene expression is not yet fully understood, but the roles of histone acetylation and methylation, pioneer transcription factors, and architectural proteins such as CCCTC binding factor (CTCF) and cohesin have recently attracted attention. Moreover, accumulated data suggest that E–P interactions are mechanistically involved in biophysical events, including liquid–liquid phase separation, and in biological events, including cancers. In this review, we discuss various mechanisms that regulate eukaryotic gene expression, focusing on emerging views regarding chromatin conformations that are involved in E–P interactions and factors that establish and maintain them.

## 1. Introduction

The spatial organization of the genome into transcriptionally active and silenced chromatin plays a fundamental role in the three-dimensional (3D) architecture required for the regulation of eukaryotic genes [1,2]. Genomic sequences are partitioned into one of two nuclear compartments called the A/B compartments: the A compartment is an ‘open’ state that allows for the transcription of the associated genes (euchromatin), and the B compartment is a ‘closed’ state associated with inactive genes (heterochromatin) [3]. The compacted chromatin of heterochromatin is assumed to be inaccessible to transcriptional machinery and resistant to chromatin remodeling; this condensed state is thus accepted as a major hallmark of repressed chromatin, which comprises silenced genes [4]. Heterochromatin is further divided into two types, constitutive and facultative. In constitutive heterochromatin, repetitive sequences such as pericentromeric regions are organized into silent nuclear compartments that are highly enriched in trimethylated histone H3 lysine 9 (H3K9me3) and methylated DNA [5]. In contrast, facultative heterochromatin consists of transcriptionally silent regions that become activated depending on the context [6].

In one key mechanism, gene transcription is regulated through chromatin loops that form between promoters and various regulatory elements, including enhancers [3,7]. Enhancers are *cis*-elements that contain diverse DNA sequences to which various transcription factors (TFs) and transcriptional co-activators bind and that are enriched with various histone modifications that facilitate gene transcription [8,9,10]. In our review, we focus on the contributions of enhancers and their interaction with promoters (another type of *cis*-element) to transcriptional control.

## 2. Histone Modifications Involved in E–P Interactions

Histone modification is widely used as a means to classify enhancers according to their activity: H3K4me1 and the binding of trithorax-related mixed lineage leukemia (MLL) complex define primed or active enhancers; H3K27me3 is a key marker of poised or inactive enhancers; histone H3 lysine 27 acetylation (H3K27ac) is a hallmark of transcriptionally active enhancers (Figure 1) [11,12,13,14]. Recent trends highlight that rather than defining active enhancers with H3K27ac, different histone acetylation marks, such as simultaneous acetylation of histone H4 at both K5 and K8 (H4K5acK8ac) [15], Das et al., [submitted], histone H2B N-terminus multisite lysine (e.g., K5, K12, K16, and K20) acetylation (H2BNTac [16]), H3K122ac [17], and H4K16ac [18], to define active enhancers are emerging. A large number of histone modifications have been implicated in gene transcription, where H3K4me3 has been associated with gene promoter regions [19]. Table 1 summarizes the most well-characterized histone modifications and their involvement in E–P interactions.

Hyperacetylation of histone H4 at its N-terminal tail is essential for normal spermatogenesis [20,21] and occurs in several cancers, including nuclear protein in testis (NUT) midline carcinoma [22]. The bromodomain (BRD) and extra-terminal domain (BET) proteins, including BRD4, preferentially bind to histone H4 tails containing multiple acetylations within 1 to 5 amino acids, e.g., H4K5acK8ac, and only rarely bind to mono-acetylated histone species, including H3K27ac [23,24,25]. Although BRD4 primarily binds to multi-acetylated histone H4, most previous studies of BRD4 have focused on H3K27ac. BET proteins, including BRD4, have long been associated with large-scale control of the nuclear structure and higher-order chromatin organization [26,27]. For example, BRD4-NUT is an oncogenic fusion protein that drives NUT carcinoma [28]. NUT is normally expressed in post-meiotic spermatogenic cells, where it interacts with p300 and triggers genome-wide histone hyperacetylation [21]. Such interactions among BRD4-NUT, p300, and histone hyperacetylation lead to the formation of hyperacetylated nuclear ‘foci,’ corresponding to large chromosomal ‘megadomains’ (100 kb to 2 Mb) [22], which involve long-range acetylation-dependent inter- and intrachromosomal interactions [29]. This example illustrates how histone hyperacetylation (e.g., H4K5acK8ac, H2BNTac), targeted by readers or writers of BRD-containing proteins, facilitates long-range contact between enhancers and promoters and controls gene expression and how dysregulation of this process leads to disease, including cancer.

**Table 1 cancers-14-05404-t001:** Summary of histone modifications and their putative role in enhancer–promoter (E–P) interactions and transcription.

Histone Modification	Putative Role in E–P Interactions and Transcription	Most Enriched Region	Reference
H3K4me3	Activation	Promoters, bivalent promoters	[19]
H3K4me1	Activation	Enhancers	[14]
H3K27ac	Activation	Enhancers, promoters	[11,12,13,14]
H4K5acK8ac	Activation	Enhancers, promoters	[15]
H4K16ac	Activation	Enhancers	[18]
H2BNTac (K5, K12, K16, K20)	Activation	Enhancers	[16]
H3K27me3	Repression	Bivalent Promoters, poised enhancers	[30,31,32,33,34]

H3K27me3 is present at high levels in CpG islands, which are associated with the promoters of developmental genes in mammals [30,31]. In addition, along with H3K4me1, H3K27me3 is plentiful in poised enhancers (PEs), which become activated in human and mouse embryonic stem cells once these PEs are ‘marked’ with acetylation at H3K27 during the activation of associated genes [14]. In most cases, PEs are linked to developmental genes that are inactive in embryonic stem cells (ESCs) and that are expressed upon differentiation. Therefore, the PE chromatin signature is proposed to bookmark associated genes spatiotemporally and facilitate their activation once appropriate differentiation signals arise (Figure 1). Accordingly, loss of polycomb repressive complex (PRC) 2, which catalyzes the methylation of H3K27, induces a genome-wide redistribution of H3K27ac marking and activation of PEs [32]. The functional role of the polycomb-repressed or -poised state of enhancers remains unknown. Several studies suggest that polycomb-repressed enhancers may convert these regions to silencers [33,34], whereas PEs may be associated with the rapid activation of genes [35].

How do specific enhancers locate and contact promoters to accomplish transcriptional regulation? Remarkable progress has been achieved regarding the molecular principles and combinatorial logic underlying these processes [3,7], but chromatin-mediated E–P interactions remain incompletely understood. In general, chromosomal interactions create microenvironments, such as topologically associating domains (TADs, highly-interacting genomic regions defined as the population-level contact-frequency domains of higher interaction frequency within a region than between regions, [36]) and nuclear compartments that are characterized by the clustering of similar epigenetic marks, which constrain the search space and thereby increase the likelihood of interactions between enhancers and promoters in the genome [37]. In one model [7], interdependent layers of regulatory control cooperate to establish and maintain E–P interactions for robust cell-type–specific gene expression. Transcriptionally favorable *cis*-regulatory elements contain TFs, RNA polymerase II, and chromatin-remodeling and histone-modifying enzymes, whose combined activities induce chromatin accessibility. Specifically, proteins bound to enhancers and promoters may interact with each other non-randomly and preferentially [38,39], leading to cooperations that influence transcriptional regulation [7]. Chromatin loops that are marked with high levels of H3K27ac and low levels of H3K27me3 tend to change upon perturbation of PRC2, thus providing evidence that histone modification can alter the overall genomic architecture [40]. In cancer cells, H3K27ac dynamics modulate the interaction frequency between regulatory regions and can lead to allele-specific chromatin configurations that sustain oncogene expression [41]. Here, we review recent progress regarding how active or permissive histone modification alters chromatin interactions by repositioning enhancer regions or by changing the contact between local enhancers and promoters. We include specific examples to illustrate how the perturbation of E–P interactions due to genome editing, controls the expression of the genes involved.

## 3. Factors Regulating E–P Interactions

How is the genome regulatory domain formed that facilitates E–P interactions? Several mechanisms participate and involve TFs (particularly pioneer factors) at enhancers to mediate chromatin remodeling [42]; loop extrusion due to architectural proteins including CCCTC binding factor (CTCF) and cohesin [43,44]; the formation of liquid–liquid phase separation (LLPS [45,46], and the dynamic nature of open (permissive) chromatin which establishes such regulatory domains (Figure 2). Enhancers consist of clusters of TFs and their binding sites [47], and enhancer sites at closed chromatin are constrained due to the additional presence of tightly packaged nucleosomes, the basic unit of chromatin consisting of a segment of DNA around a histone octamer (Figure 1) [48,49]. The chromatin status, including its histone modification and DNA methylation, which allows the binding of TFs and architectural proteins may facilitate enhancer-mediated chromatin looping. Importantly, histone tail modifications could provide an accurate readout of enhancer activity, and the causal interactions between regulatory elements, such as enhancer–enhancer and enhancer–promoter relationships, can influence chromatin accessibility over both long (~500 kb) and short (<20 kb) distances [50].

## 4. TFs and Pioneer TFs Involved in E–P Communication

The combined activities of TFs, RNA polymerase II, and chromatin-remodeling and histone-modifying enzymes generate cell type-specific accessible chromatin regions; these factors also create marks that serve as binding sites for effector proteins [7]. These types of regulatory elements typically participate in long-range interactions. Although the gene transcriptional program depends on the specific TFs bound and the epigenetic states of promoters and enhancers, the physical interaction between enhancers and promoters creates an accessible chromatin conformation that may be sufficient to activate gene transcription [51]. The subset of TFs called ‘pioneer factors’ is uniquely capable of binding to histone-wrapped DNA, establishing accessible chromatin conformations and facilitating the binding of additional TFs (Figure 1) [52]. In addition, pioneer factors can target DNA on the surface of nucleosomes, allowing these factors to bind regions of chromatin that are inaccessible to other TFs [42]. Such properties enable pioneer factors to recruit cooperative TFs, which might otherwise be constrained in inaccessible chromatin due to the packing of counterpart enhancers in nucleosomes. Several pioneer factors, including FOXA, SOX2, and SOX11, open chromatin by displacing histones [48,49].

The binding of pioneer factors to DNA sequence motifs is context-dependent and cell-type–specific. For example, in several human cell lines, including liver carcinoma (Hep2), lung carcinoma (A540), and ESC-derived endoderm, FOXA2 show cell-type–specific binding: only 6% to 14% of FOXA2 binding occurs in its recognition motifs in the mentioned cell types [53]. Cell-type–specific differences in pioneer factor occupancy occur largely at enhancers [54,55]. Chromatin accessibility at enhancers is correlated more closely with tissue-specific gene expression than with the accessibility of promoters, which are often accessible even in tissues in which the gene they regulate is not expressed [56].

In addition, several other proteins, including yin and yang 1 (YY1), zinc finger protein 143 (ZNF143), and Myc-associated zinc finger protein (MAZ), may contribute to the organization of chromosomal architecture, especially of E–P interactions in mammals (Figure 2). YY1 is an evolutionarily conserved TF that is ubiquitously expressed in mammals and thus participates in many biological processes [57,58]. YY1 protein forms homodimers, which facilitate long-range E–P interaction through the binding of YY1 with its consensus sequence motif in DNA. Consistent with a potential role of YY1 in E–P interaction, genome-wide analyses have demonstrated that YY1 predominantly associates with enhancers and promoters [59]. YY1 is overexpressed and consequently correlates with poor prognosis in many types of cancer, and YY1 regulates a cohort of cancer-related genes [60]. Therefore, investigating the causal relationship among overexpression of YY1, E–P contacts, and gene expression is crucial.

Like YY1, ZNF143 is a regulator of E–P contact [61]. Conditional knockout of *Znf143* in adult hematopoietic stem and progenitor cells leads to the loss of CTCF binding on promoters and enhancers and induces changes in gene expression [62]. Importantly, CTCF-bound E–P interactions are disrupted in *Znf143*-knockout hematopoietic stem and progenitor cells, whereas TAD formation and compartmentalization in the knockout cells are not affected [62]. This finding suggests that ZNF143 is a CTCF regulator that mediates CTCF–DNA binding in promoters, enhancers, and associated E–P contacts. Similarly, recent CRISPR library screening identified MAZ as a cofactor of CTCF in insulator elements, although whether MAZ is required for E–P interaction was not studied [63].

## 5. The Architectural Proteins CTCF and Cohesin

CTCF mediates chromatin looping by associating with the cohesin complex [64,65]. The cohesin complex forms a ring-like structure, which facilitates chromatin looping both *in cis-* and *in trans-* (in sister chromatid) fashion [66]. CTCF plays a vital role in stabilizing CTCF–cohesin-associated chromatin loops [44]. In addition, CTCF independently mediates chromatin loop formation at the nucleolar surface [67], and CTCF binding sites near the *β-globin* locus could form a chromatin loop [68]. In addition to mediating specific E–P loops, the cooperative action of cohesin and CTCF has emerged as a central organizer of global chromosomal architecture through a mechanism termed ‘loop extrusion,’ in which the megabase-scale segregated domain forms a TAD (Figure 3) [69]. According to the loop-extrusion mechanism, cohesin in interphase nuclei continuously extrudes chromosomal loops until it encounters a boundary that prevents further extrusion. Such barriers arise when CTCF binds to its sequence motif located in divergent orientations at either end of the loop [70,71], thus forming an extended loop domain (i.e., TAD), consistent with CTCF’s established role as an insulator-associated protein. Notably, TADs correspond to only one hierarchical level of such ‘loop domains,’ with sub-TADs and insulated neighborhoods potentially corresponding to other levels [72,73]. However, the level of the hierarchy corresponding to TADs has been suggested to be ‘functionally privileged,’ given that their boundary motifs are most abundant in CTCF binding and show the greatest conservation across cell types [74].

## 6. BRD4 and the Mediator Complex

The transcriptional co-factor BRD4 facilitates the recruitment of Mediator (a multiprotein complex involved in the transcriptional regulation by RNA polymerase II [75]) to active enhancers, thus leading to important roles in pluripotency and cancer [76,77]. In addition, Mediator acts as a looping factor with CTCF and cohesin [78]. At active enhancers, Mediator participates with other co-activators to form the transcription pre-initiation complex and facilitates the recruitment of RNA polymerase II (Figure 2) [79]. However, the use of chemical inhibitors decreased the binding of BRD4 (and consequently of Mediator) at enhancers, leading to a dramatic and rapid downregulation of gene expression, but E–P looping structures remained stable and intact [80]. In addition, independent removal of Mediator from chromatin does not markedly affect E–P interactions but rather causes widespread transcriptional changes and cell-cycle arrest [77]. In contrast, in a subset of genes, the presence of CTCF and cohesin in combination with low levels of BRD4, Mediator, and other TFs facilitates E–P interactions [80]. We surmise that BRD4 and Mediator are not independently essential for chromatin looping, but rather that this looping occurs due to the cooperative binding of BRD4 and Mediator with CTCF and cohesin.

## 7. Polycomb Group (PcG) Complexes and LIM Domain Binding Proteins

PcG complexes and LIM domain binding proteins may contribute to E–P interactions via homotypic contacts (i.e., the force driving chromatin-binding proteins to preferentially self-associate; [37]). PcG proteins contribute to the formation of facultative heterochromatin and mediate gene silencing through at least two distinct protein complexes, PRC1 and PRC2 (Table 2) [81,82]. PRC2 mediates H3K27me3 [83,84], and PRC1 mediates the mono-ubiquitylation of histone H2AK119 (H2AK119u1) via the RING finger E3 ligases RING1A and B [85]. Independent of H2AK119u1, the sterile α motif (SAM) domain of the PRC1 component, poly-homeotic-like proteins 1, 2, and likely 3 facilitate the condensation of target gene loci through polymerization of the SAM domain; this polymerization in turn contributes to clustering of PRC1 proteins and the LLPS-like formation of the ‘polycomb body,’ which has been linked with transcriptional repression of developmental genes (Figure 2) [86,87,88]. Indeed, knockout of PRC1 in ESCs obliterates homotypic contacts, including P–P interactions, thus causing transcriptional upregulation of PRC1 target genes. In addition to modulating P-P contacts, PcG proteins can mediate the formation of chromatin loops between various regulatory elements, including promoters, enhancers, and silencers [89,90]. Therefore, PcG might contribute not only to transcriptional repression of downstream target genes but also to prospective activation, likely via chromatin loops that constrain inactive/poised enhancers and promoters in close proximity [91] and that thus facilitate subsequent activation-associated interactions to activate transcription (Figure 2).

LIM domain binding protein 1 (LDB1) is a nuclear adapter protein that interacts with the C-terminal LIM domain in numerous LIM domain–containing TFs [92]. In addition, LDB1 contains a conserved, N-terminal dimerization domain through which the protein can form homodimers and higher-order oligomers [93,94,95]. Through such homotypic contacts, LDB1 facilitates E–P interactions and is required for the activation or repression of various genes in multiple cell lineages, including neuronal, cardiac, and hematopoietic [51,95].

## 8. E–P Interactions in LLPS

LLPS is the formation of membrane-less biomolecular condensates and occurs when overall protein(s) content increases to a critically high point, leading to the formation of phases according to the concentration (i.e., high or low) of the associated proteins [45,46]. The weak protein–protein interactions involving intrinsically disordered regions of the involved proteins facilitate the formation of LLPS [96]. In that regard, the intrinsically disordered regions of TFs when they are plentiful [96] and the intrinsically disordered nucleosome tails associated with abundant histones [97,98] might promote LLPS. In addition, these chromatin-associated TFs and histones may coordinate with various cofactors, including BRD4 and Mediator complexes [96,99], via weak, multivalent interactions. In that situation, super-enhancers (SEs; i.e., clusters of active enhancers) may lead to LLPS, where the histones and other factors associated with SEs may be present in amounts above the threshold required for LLPS formation [46,100].

LLPS may provide a mechanistic explanation of the classic ‘transcription factory’ model [101], whereby specific activators or repressors are assembled in high concentration at appropriate genomic loci [102]. Importantly, local chromatin features, including marks indicating active (H3K27ac; [98]) or repressed (HP1 and H3K27me3; [97] chromatin, the presence of linker histone H1 proteins, and the associated TFs [103] cooperate in forming or dissolving phase-separated condensates (Figure 2). In addition, newly identified marks of active enhancers, such as histone H4 hyperacetylation (H4K5acK8ac; Das et al., submitted) and H2BNTac [16], may contribute to the formation of phase-separated condensates. As directed by local chromatin features (e.g., in SEs), such phase-separated condensates could mediate systematic loading of the transcriptional machinery to active gene promoters [102], thus facilitating the E–P interactions. Furthermore, the recently coined ‘enhancer hub’ associated with CTCF and cohesin [104] may have similar properties to LLPS [105], where various interacting factors form the regulatory elements that achieve a dynamic system of transcriptional regulation. Namely, the enhancer hub represents a dynamic and heterogenous network of multivalent interactions where multiple enhancers target a single promoter or interconnecting enhancers target more than one promoter that regulate the expression of spatially connected genes [106].

## 9. E–P Interactions Associated with Chromatin and Diseases

Now that we have reviewed the factors and mechanisms that influence the formation of E–P interactions, we focus our discussion on the role of chromatin-mediated E–P contact in transcriptional control. In permissive chromatin, the contact between enhancer and promoter regions appears to be sufficient to induce transcription in the absence of lineage-specific transcription factors [51]. For example, forced chromatin looping due to induced contact between the mouse β-globin (*Hbb*) promoter and its enhancer led to strong transcriptional activation of *Hbb*, even in the absence of the lineage-specific transcription factor GATA1 [51], thus directly demonstrating that E–P interactions by themselves can induce transcription. The manipulation of chromatin looping might have potential therapeutic applications when the enhancer regions of associated genes are known and accessible, e.g., the γ-globin gene (*HBG*) involved in β-thalassemia and sickle cell anemia [51,107]. However, E–P interactions do not demonstrate a binary state, such that the interaction either occurs or does not. Rather, chromatin-associated E–P interactions can assume diverse conformations with varying abilities to influence gene expression [108].

In this section, we discuss how repositioning just a few histone modifications—especially in cancer cells—changes local interactions between enhancer and promoter regions. Increased levels or breadth of H3K27ac enrichment at enhancer regions has been associated with oncogenic enhancer activity in human colorectal cancer cells, even in the absence of chromosomal alterations [109]. Although H3K27ac is widely used to predict enhancer activity, only 12% of H3K27ac ChIPseq peaks functioned as enhancers in a massive parallel reporter gene assay in human ESCs [110]. In addition, H3K27ac is dispensable for transcription, even of genes associated with enhancers or SEs [111], and is not essential for the de novo activation of genes during the cell-fate transition of mouse ESCs to epiblast-like cells [112]. Such evidence suggests that *cis*-regulatory sequences devoid of classic enhancer chromatin marks might have the potential to perform enhancer-like functions and provide the E–P contact necessary for transcription. For example, in mouse ESCs, a class of non-canonical enhancers marked by H3K122ac—but lacking H3K27ac—controls the expression of target genes, including key pluripotency genes such as *Tbx3* and *Sox2* [17]. Using a CRISPR-Cas9–mediated approach, our group revealed that deletion of the H4K5acK8ac-preferred SEs associated with *MYCN* and *NFIC* reduced their expression in human glioblastoma stem-like cells and diminished these cells’ stem-like properties (Das et al., submitted). Together, these data suggest that alternative sites for histone acetylation other than H3K27ac—modified by either p300/CBP or other enzymes—might maintain E–P contact and thus lead to gene activation.

Regarding heterochromatin, H3K27me3-rich regions consist of silencers that can repress target genes, such as tumor suppressors in human cancer cells, either in proximity or through long-range chromatin interactions [40]. CRISPR-mediated knockout of H3K27me3-enriched silencer regions associated with IGF2 changes in chromatin loops, particularly distant loops, histone modifications, and cell phenotypes. In this context, loops with high levels of H3K27ac but low H3K27me3 tend to change in conformation, thus providing evidence that histone modification does in fact influence the overall genomic architecture. Given that PRC1 and PRC2 are necessary to maintain the chromatin interaction landscape [113], the inhibition of EZH2, a catalytic subunit of PRC2, due to treatment with GSK343 changes the chromatin interaction, reduces the H3K27me3 level and induces the up-regulation of the associated gene [40]. Alternatively, heterochromatin can be targeted through acidic TF activation domains or with histone deacetylase inhibitors; these techniques counteract heterochromatinization even in the absence of active transcription [114,115,116]. The mechanism through which H3K27me3-enriched silencers interact with tumor suppressor genes has yet to be fully resolved; however, perturbing such regions may have therapeutic potential in cancer by activating associated tumor suppressor genes.

Along with CTCF and cohesin, post-translational modification of histones is necessary to determine the 3D structural organization of chromatin (Figure 3). The enrichment of particular histone marks in various regions of chromatin is associated with their compartmentalization, where active and inactive histone marks, respectively, define the A and B compartments of chromatin [117,118]. The allocation of chromatin into its A and B compartments is independent of CTCF and cohesin [119,120]. In human cancer cells, histone hyperacetylation driven by a fusion oncoprotein (BRD4-NUT) facilitates the formation of a newly recognized chromatin interaction with a unique sub-compartment [29].

In human leukemic cells, the increased accumulation or distribution of H3K27ac at specific enhancers serves to ‘zip’ together these regions to increase the E–P contact frequency and the number of interacting regions of target gene promoters, thus influencing oncogene expression [41]. Importantly, the modification of H3K27ac levels due to chromosomal rearrangement, stalling of cell differentiation, or pharmacological or genetic perturbation of histone acetyltransferase activity alters the frequency of interaction between these chromatin structures and can modulate gene expression [41]. BRD4 binds to acetylated histones and NUT recruits p300, consequently leading to a broad distribution of BRD4-NUT and histone acetylation across long stretches of chromatin in human NUT midline carcinoma [22]. Blocking the catalytic activity of p300 induces a genome-wide reduction in H3K27ac leading to an overall loss of chromatin interaction and targeted activity of dCas9–p300 in the enhancer region and inducing the formation of chromatin interactions between the enhancer and proximal promoter region, thereby altering gene expression in leukemic cells [41]. Whereas TAD dysfunction due to boundary misregulation occurs in several cancers, including glioma [121] and T-cell acute lymphoblastic leukemia [122], oncogenic hyperacetylated BRD4-NUT results in pathological long-range interactions within a novel nuclear sub-compartment in NUT carcinoma [29]. However, the involvement of H3K27ac in 3D chromatin architecture should be considered with caution because there is a report suggesting that catalytic inhibition of p300 perturbs transcription but does not affect the 3D organization of chromatin [123].

## 10. Conclusions

The high levels of context-specific histone marks, TFs, and pioneer TFs at enhancers relative to promoters proclaim that E–P interactions are likely important to the expression of many genes. In addition, the expression level of a gene and the number of promoters interacting with enhancers are positively correlated [124,125], thus supporting the role of E–P contact in target gene expression. Some reports indicate that physical contact between an enhancer and promoter is not intrinsically sufficient for transcription and is not dependent on high levels of associated transcriptional co-activators, such as BRD4 or Mediator [80]; these findings suggest that context specificity regarding transcriptional coactivators and histone modifications is crucial to E–P interactions and associated gene expression. Furthermore, we mentioned earlier that E–P interactions are highly dynamic and are regulated not only by CTCF and cohesion but also by other factors, including YY1, ZNF143, MAZ, PcG, and LBD1 among others. It is critically important to elucidate the spatiotemporal requirement for each factor and the redundancy in mediating E–P interactions.

We surmise that, through the combined action of various histone acetyltransferases, diverse histone acetylations, including H4K5acK8ac and H2BNTac, create an acetylation-dependent condensate and recruit regulatory factors to enhancers to establish their interaction with a specific promoter, resulting in transcriptional activation. Given that histone modifications can alter the genomic architecture, the conformation of chromatin loops with variable active and permissive histone marks changes, leading to either a gain or loss of E–P interactions and subsequent up- or down-regulation of the expression of the associated gene. To overcome the difficulty in determining the causal order among architectural proteins, TFs, histone marks, and gene expression with E–P contact, CRISPR-Cas9 genome editing might be applied to assess the role of each factor, especially that of enriched histone marks in maintaining E–P contact. The functional relationship between E–P contact and gene expression is rather complex in terms of chromatin structure, and further mechanistic studies are required to explain this dependence. The integration of rapidly advancing computational models, locus-specific imaging, and functional perturbation through CRISPR–Cas9 genome editing, inducible degron systems, and epigenetic inhibitors hold great promise in addressing such questions.

## Figures and Tables

**Figure 1 cancers-14-05404-f001:**
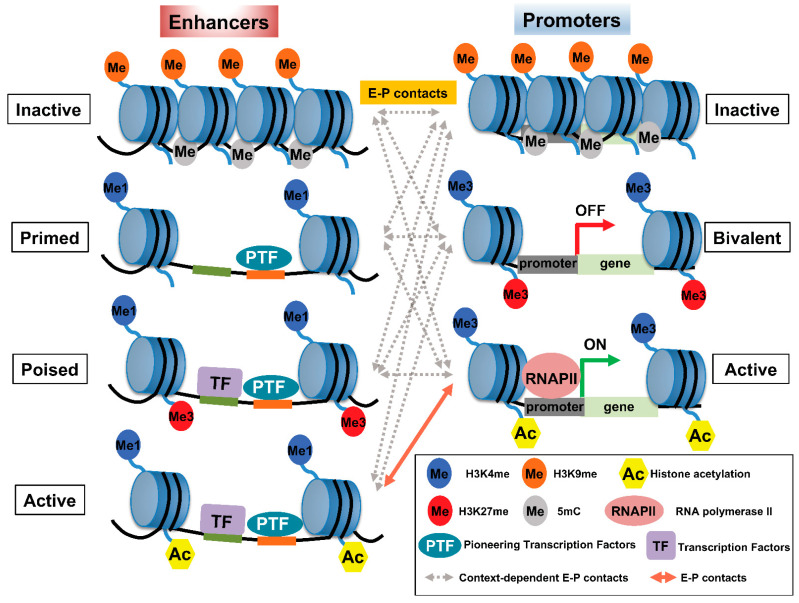
Pattern of histone modifications defines enhancer–promoter (E–P) states. (Left panel) The pattern of histone modifications in various enhancer states. Inactive enhancers are characterized by high nucleosome density and enrichment of H3K9me2/3 and DNA methylation. At primed enhancers, pioneering TFs (PTF) bind to their target sites and recruit MLL3/4 complexes to decorate H3K4me1. PTF binding is accompanied by nucleosome remodeling. H3K27me3 may contribute to the poised status of an enhancer to prevent premature activation. In addition, poised enhancers may allow lineage-specific TFs (LTF) to bind and recruit histone acetyltransferases, such as p300/CBP, and H3K27 demethylase(s) to prepare the enhancer for rapid activation. Active enhancers contain not only H3K27ac—the hallmark of an active enhancer—but also H4K5acK8ac, H2BNTac, and potentially acetylation at various residues of other histones. (Right panel) The pattern of histone modifications in various promoter states. Promoter activity is highly correlated with enhancer activity and the associated epigenetic landscape. Inactive promoters carry high levels of H3K9me2/3 and DNA methylation. The vast majority of each promoter is marked by H3K4me3, whereas other parts of the promoters are marked by H3K27me3, thus constituting so-called ‘bivalent promoters,’ which are repressed by polycomb group complexes. In comparison, active promoters gain histone acetylation marks. The epigenetic landscape of enhancers and promoters can be variably altered in response to external signals. However, the timing of and how E–P interactions facilitate gene expression is not well understood and may involve context-dependent interactions among enhancers and promoters at various states.

**Figure 2 cancers-14-05404-f002:**
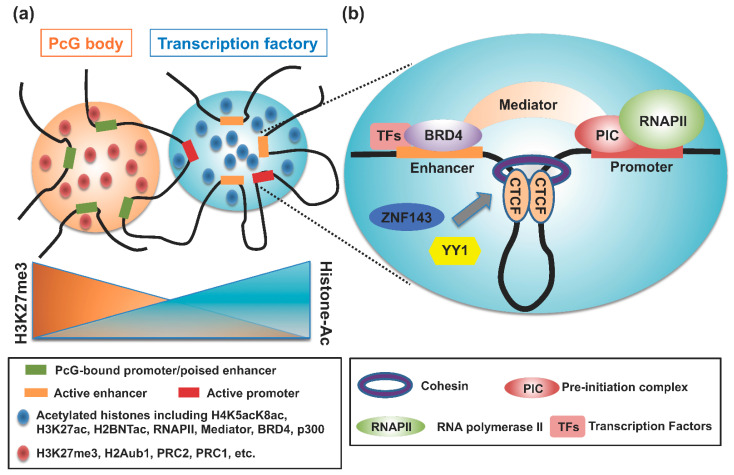
The central role of genome organization in transcription. (**a**) The formation of PcG bodies due to liquid–liquid phase separation (LLPS) organizes PcG-bound chromatin that is enriched in H3K27me3 and H2Aub1. PcG bodies may contain poised enhancers that are marked with H3K27me3 and H3K4me1. At actively transcribed loci, the transcription factory may undergo LLPS, to assemble cell-specific activators, including active histone marks (e.g., H3K27ac, H4K5acK8ac, H2BNTac) and associated transcription factors (TFs), in high concentrations. In addition, super-enhancers may mediate systematic loading of the transcriptional machinery to active gene promoters, and the recently coined ‘enhancer hub’ associated with CTCF and cohesin may have similar properties to LLPS. (**b**) E–P interactions mediated via multiple factors ensure active transcription.

**Figure 3 cancers-14-05404-f003:**
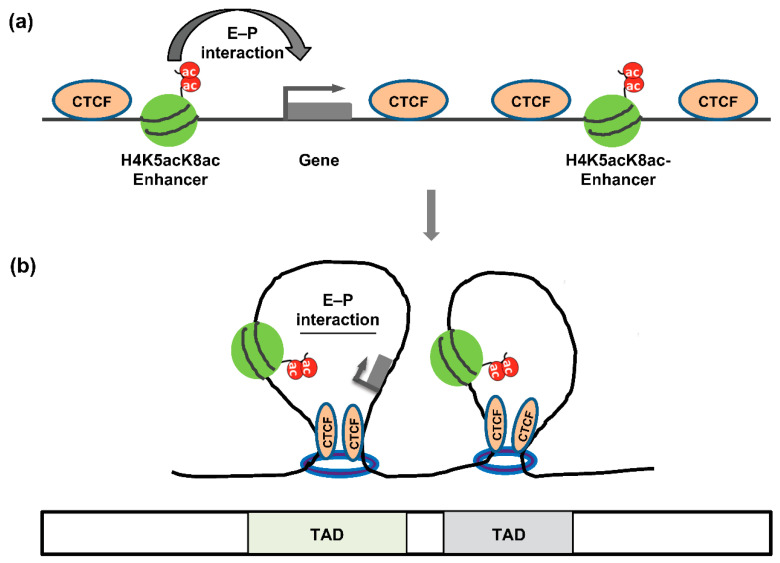
Schematic of E–P interaction and TAD into genome organization. (**a**) A hypothetical E–P interaction. An enhancer enriched with histone modification, (*i.e.,* H4K5acK8ac) drives the expression of the associated gene by forming the E–P interaction. CTCF enrichments are shown that mediate the formation of TAD (**b**) TADs form the genome organization. An E–P interaction occurs within the TAD, and regions within the TAD are more likely to interact each other than regions outside the TAD. Cohesin is indicated by blue rings.

**Table 2 cancers-14-05404-t002:** The core components of PRC1 and PRC2 and their functions in chromatin organization [81,82].

Complex	Subunit	Domain	Potential Function
PRC1	RING1A/RING1B	RING finger	H2AK119 ubiquitination
	PCGF1-6	RING finger	Enhancer of RING1A/RING1B activity
	PHC1-3	SAM	Oligomerization
	CBX2,-4,-6,-7,-8	Chromodomain	H3K27me3 binding
PRC2	EZH1/EZH2	SET	H3K27 methylation
	EED	WD40	H3K27me3 binding
	SUZ12	Zinc finger	PRC2 stability
	RBBP4/RBBP7	WD40	Nucleosome binding

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
