# Peer review of "Factors and Mechanisms That Influence Chromatin-Mediated Enhancer–Promoter Interactions and Transcriptional Regulation"

_cancers, 2022, doi:10.3390/cancers14215404_

Round 1

Reviewer 1 Report

This review focuses on the mechanisms involved in the interaction between enhancers and promoters. The review has an interesting approach to these mechanisms and clearly states them.

These are some of specific recommendations to improve the review.

In your introduction please, define histones and nucleosome, define PTMs and what factors will recognize them, for example define the domains that will recognize the PTMs of interest to this review.

Define mediator.

Figure 1: Define epigenetics in this context.

Could you please confirm in the figure caption that the H3K27me3 is in a different histone tail than H3K4me3 and H3K4me however in the same nucleosome? Explain the arrows.

Figure 1 and do not need title "Figure 1" and “2”

Define H2BNTac

Define TADs

Throughout the text please specify if the cited examples are in humans, mammalian or in vitro experiments.

Author Response

Reviewer-1:

This review focuses on the mechanisms involved in the interaction between enhancers and promoters. The review has an interesting approach to these mechanisms and clearly states them.

These are some of specific recommendations to improve the review.

In your introduction please, define histones and nucleosome, define PTMs and what factors will recognize them, for example define the domains that will recognize the PTMs of interest to this review.

Response: We thank the reviewer for the suggestion. In the revised manuscript, we have defined nucleosome in line 134 and prepared a table listing histone modifications and their functions in enhancers and promoters. We have also explained the key factors such as BET proteins that recognize acetylated histones in lines 71-77.

Define mediator.

Response: We have explained in lines 203-204, with specific reference; or “a multiprotein complex involved in the transcriptional regulation by  RNA polymerase II”.

Figure 1: Define epigenetics in this context.

Response: We have amended the figure title and changed it accordingly in the figure legend.

Could you please confirm in the figure caption that the H3K27me3 is in a different histone tail than H3K4me3 and H3K4me however in the same nucleosome? Explain the arrows.

Response: We thank the reviewer for the comment. There is still an argument as to whether two histone modifications, (i.e., H3K27me3 and H3K4me3) are present in different histone tails in the same nucleosome. This is because the resolution is not sensitive enough to distinguish the issue experimentally with ChIP/ ChIP-seq or similar types of experiments. However, bivalency was intensively discussed in the review (Figure 3 in Reference 31, Voigt P. et al 2013 Genes and development), where they argued that bivalent domains feature nucleosomes that carry H3K4me3 and H3K27me3 on opposite H3 copies. Therefore, we believe that the cartoon shown in Figure 1 is correct, at least as currently interpreted. We have cited the above-mentioned review in the revised version.

Figure 1 and do not need title "Figure 1" and “2”

Response: We have corrected all figures.

Define H2BNTac

Response: We have mentioned the multisite of lysine, i.e., K5, K12, K16, and K20 acetylation of H2B N-terminus, in line 66, as “histone H2B N-terminus multisite lysine (e.g., K5, K12, K16, and K20) acetylation (H2BNTac; [16]).”

Define TADs

Response: We have explained this in lines 105-107.

Throughout the text please specify if the cited examples are in humans, mammalian or in vitro experiments.

Response: We have now specified whether the cited examples are in mammalian or in vitro experiments, especially in the section on “E–P interactions associated with chromatin and diseases.”

Reviewer 2 Report

In this review manuscript, Ito et al. summarize information regarding chromatin conformations that are involved in E–P interactions (enhancers and promoters physically interact), and factors that establish and maintain them. In general, it is a well-written review manuscript. Here are my comments:

1. The introduction section seems too long. The detailed information regarding histone modifications in the introduction section can be separated as an individual section or merged with other sections.

2. Line 154: please add reference(s) regrading ‘enhancer hub’. It is better to add 1-2 sentences about basic information of ‘enhancer hub’

3. Line 155: what is “(right panel)”?

4. Line 240: please clarify/correct “via homotypic contacts (i.e., self-association of proteins with similar properties;”

5. It is better to divide the section “E–P interactions associated with chromatin and diseases” into two sections, with one focusing on chromatin in physiological conditions (e. g. ESC), and the other focusing on diseases (e. g. cancer).

6. Line 347-348; what is “A and B compartments of chromatin”?

Author Response

Reviewer-2:

In this review manuscript, Ito et al. summarize information regarding chromatin conformations that are involved in E–P interactions (enhancers and promoters physically interact), and factors that establish and maintain them. In general, it is a well-written review manuscript. Here are my comments:

  1. The introduction section seems too long. The detailed information regarding histone modifications in the introduction section can be separated as an individual section or merged with other sections.

Response: We have rearranged the introduction and created a separate section as “Histone modifications involved in E-P interactions.”

  1. Line 154: please add reference(s) regrading ‘enhancer hub’. It is better to add 1-2 sentences about basic information of ‘enhancer hub’

Response: As suggested, we have added a reference [106] to lines 264 to 269 to explain in more detail as follows: “Namely, enhancer hub represents a dynamic and heterogenous network of multivalent interactions where multiple enhancers target a single promoter or interconnecting enhancers target more than one promoter that regulates(?) the expression of spatially connected genes [106].”

  1. Line 155: what is “(right panel)”?

Response: For clarity, we have modified and divided the figure into (a) and (b).

  1. Line 240: please clarify/correct “via homotypic contacts (i.e., self-association of proteins with similar properties;”

Response: We have corrected the description to ‘homotypic contacts (i.e., the force driving chromatin binding proteins to preferentially self-associate; [37])’

It is better to divide the section “E–P interactions associated with chromatin and diseases” into two sections, with one focusing on chromatin in physiological conditions (e. g. ESC), and the other focusing on diseases (e. g. cancer).

Response: In this section, we mainly focus on the E–P interactions in diseases, especially cancer. For the smooth discussion, we have added only a few cited examples in physiological conditions where chromatin-mediated E–P interactions are associated with gene transcription in mouse ESCs. Since there are not so many cited examples have been added in physiological conditions, we keep the discussion together in this section.

  1. Line 347-348; what is “A and B compartments of chromatin”?

Response: We have defined the A and B compartments in the introduction (Lines 42-44).

Reviewer 3 Report

The manuscript is well written and provides informative details about the enhancer - promoter interaction for transcriptional regulation. I would suggest few minor points which should be considered in this manuscript. 

- The authors should add few tables in the manuscript for simplicity:

1. One table provides information about different types of histone marks as described in line 46 - 54 and what it means in terms of E/P activation or silenced?

2. One table should be added to talk about the different components of PRC1 & PRC2 and their function/roles.

3. Another table should be considered to provide the list of major chromatin remodelers  or pioneer TFs and what cell type they correspond with their function.

4. It would be good if authors adds an another figure to show TADs with chromatin looping and cohesin rings. 

Author Response

Reviewer-3:

The manuscript is well written and provides informative details about the enhancer - promoter interaction for transcriptional regulation. I would suggest few minor points which should be considered in this manuscript. 

- The authors should add few tables in the manuscript for simplicity:

  1. One table provides information about different types of histone marks as described in line 46 - 54 and what it means in terms of E/P activation or silenced?

Response: We have added Table 1 showing a summary of histone modifications and their role in enhancer–promoter (E–P) interactions and transcription.

  1. One table should be added to talk about the different components of PRC1 & PRC2 and their function/roles.

Response: We have added Table 2 showing the core components of PRC1 and PRC2 and their functions in chromatin organization.

  1. Another table should be considered to provide the list of major chromatin remodelers  or pioneer TFs and what cell type they correspond with their function.

Response: We thank the reviewer for the suggestion to prepare the table listing chromatin remodelers or pioneer TF. In this review, however, we focus on chromatin modifications and their putative role in enhancer-promoter contacts. Therefore, we would say that the suggested table is out of scope.

  1. It would be good if authors adds an another figure to show TADs with chromatin looping and cohesin rings. 

Response: We have added Figure 3. In the main text, ‘Figure 3’ at line 192 and line 321 have been added where the description has been matched with ‘Figure 3’.

Round 2

Reviewer 1 Report

The review has improved significantly and reads very well.